# Effects of COVID-19 and Social Distancing on Rhinovirus Infections and Asthma Exacerbations

**DOI:** 10.3390/v14112340

**Published:** 2022-10-25

**Authors:** Jordan E. Kreger, Marc B. Hershenson

**Affiliations:** Department of Pediatrics, University of Michigan Medical School, Ann Arbor, MI 48109, USA

**Keywords:** asthma, COVID-19, exacerbation, rhinovirus, SARS-CoV-2

## Abstract

Since their discovery in the 1950s, rhinoviruses (RVs) have been recognized as a major causative agent of the “common cold” and cold-like illnesses, accounting for more than 50% of upper respiratory tract infections. However, more than that, respiratory viral infections are responsible for approximately 50% of asthma exacerbations in adults and 80% in children. In addition to causing exacerbations of asthma, COPD and other chronic lung diseases, RVs have also been implicated in the pathogenesis of lower respiratory tract infections including bronchiolitis and community acquired pneumonia. Finally, early life respiratory viral infections with RV have been associated with asthma development in children. Due to the vast genetic diversity of RVs (approximately 160 known serotypes), recurrent infection is common. RV infections are generally acquired in the community with transmission occurring via inhalation of aerosols, respiratory droplets or fomites. Following the outbreak of coronavirus disease 2019 (COVID-19), exposure to RV and other respiratory viruses was significantly reduced due to social-distancing, restrictions on social gatherings, and increased hygiene protocols. In the present review, we summarize the impact of COVID-19 preventative measures on the incidence of RV infection and its sequelae.

## 1. Rhinoviruses

RVs are the major etiologic agent for the “common cold”. First isolated in the 1950s [1,2], RV infection causes symptoms such as rhinorrhea, nasal congestion, cough, sore throat, headache, and general malaise. A member of the genus *Enterovirus* within the *Picornaviridae* family, RV is a non-enveloped, positive-sense, single-stranded RNA virus with an icosahedric capsid containing a 7.2-kbp genome [3,4]. The viral genome encodes a single polyprotein which is cleaved by viral proteases to produce four capsid proteins (VP1, VP2, VP3, and VP4) and several non-structural proteins necessary for genome replication and virion assembly. VP1, VP2, and VP3 are present on the cell surface and contribute to the antigenic diversity while VP4 is internally located and forms a multimeric membrane pore, allowing extrusion of viral RNA genome into the cytoplasm [5].

There are three distinct RV species (RV-A, RV-B, and RV-C) comprising around 160 serotypes, each of which are categorized by nucleotide differences in the capsid protein regions and cellular receptor specificity. The majority of RV-A and RV-B serotypes (termed the “major group”) bind intercellular adhesion molecule 1 (ICAM-1) for host cell entry [6,7]. ICAM-1 is member of the immunoglobulin (Ig) superfamily notable for its role in promoting pro-inflammatory leukocyte migration. The remaining 12 RV-A and RV-B serotypes (termed the “minor group”) alternatively utilize low density lipoprotein receptors (LDLR) [8]. In 2007, a new species of RV, RV-C, was discovered [9,10]. Cadherin-related family member 3 (CDHR3) has recently been determined to mediate binding and replication of RV-C serotypes CDHR3 [11]. Growing evidence suggests that RV-C and RV-A may be more virulent than RV-B. Retrospective studies have reported that RV-C, followed closely behind by RV-A, was the most frequently detected RV species among pediatric intensive care unit and pediatric emergency department patients admitted for acute respiratory infections [12,13]. Additionally, RV-C was particularly prevalent in children less than three years of age admitted with higher severity respiratory infection [14], as well as children less than 16 years of age with severe wheezing and febrile illness [15]. Children with RV-C infection are more likely than those with RV-A to have underlying high-risk conditions such as asthma and to have a discharge diagnosis of asthma [16].

It was once thought that RVs did not contribute to lower respiratory tract disease. Based on part on the notion that RVs replicate more readily at 33 °C, the temperature of the nasal passage, compared to 37 °C, it was thought that RV could not infect the lower airways [17]. However, optimal temperature for replication may vary among different serotypes and viral replication at 37 °C may still produce high enough titers to initiate infection [18]. Following experimental infection, RV has been detected in the lower airways by immunostaining, PCR and in situ hybridization for positive strand viral RNA [19,20].

Recent epidemiological studies using PCR-based assays have provided insight into the impact of RVs on more severe respiratory illnesses, in particular exacerbations of asthma, COPD and other chronic respiratory diseases. PCR-based studies looking at the prevalence of virus identification among various cohorts of patients with asthma when sick and well consistently show a higher prevalence of viral infection during exacerbations. Outpatient children who are sick with asthma exacerbations show anywhere from 62–81% positivity for viral infection versus only 12–41% of children who are well [21]. Picornaviruses (primarily RV) were detected in 65% of cases, human coronaviruses (HCoVs) in 17%, influenza and parainfluenza viruses in 9% and RSV in 5% [21]. Similar results have been found in hospitalized children, adult outpatients and hospitalized adults [22,23,24,25,26,27,28,29]. The prevalence of RV-induced asthma exacerbations may vary with the time of year and prevailing viruses in the community. In one study, RV was detected in 82% of all children admitted to an emergency room for acute asthma between January and July [30]. Twenty-two to 64% of patients with COPD exacerbations have virus identified by PCR versus 12 to 19% of non-exacerbating subjects [31,32,33,34,35].

Additional information that viruses indeed cause attacks of chronic airways disease comes from an analysis of emergency department presentations for asthma and COPD over the course of a calendar year [36]. Exacerbations of asthma in children peak after school return from summer vacation (in North America, the first week of September) and around the Christmas holiday, consistent with an infectious cause. These are soon followed by asthma exacerbations in adults and COPD exacerbations in the elderly. Thus, social interactions with children drive RV infections and exacerbations of asthma and COPD in adults, similar to what has been found for severe acute respiratory syndrome coronavirus 2 (SARS-CoV-2) [37].

Finally, recent studies have implicated RV in viral bronchiolitis in infants, pneumonia and asthma development in children. RV is responsible for between 20–30% of all cases of bronchiolitis, and the most common cause in children 24 months or older [38]. RVs were identified in 29.0% of cases of community-acquired pneumonia [39]. A particular focus has been placed on the role of early life RV infections in the development of asthma. Several studies show a significant association between RV-induced wheezing in the first three years of life and subsequent development of recurrent wheezing and asthma [40,41]. A multicenter prospective study of U.S. infants hospitalized for bronchiolitis found that only infants with RV-C had a higher risk of physician-diagnosed asthma at 4 years of age [42].

While there has been a great deal of progress in efforts to understand the immunological basis of RV infection, no effective antiviral treatments have been developed (reviewed in [43]). On the other hand, new biologic therapies designed to regulate components of inflammation in severe asthma hold great promise for reducing the risks of viral infection. For example, omalizumab, a monoclonal antibody against IgE, has been shown to significantly reduce exacerbations in children with allergic asthma [44,45]. In one study, respiratory viruses were detected in 89% of exacerbations, with 81% due to RV [45]. Peripheral blood monocytes from patients who received omalizumab showed a significant increase of interferon-α generation in response to RV in vitro, suggesting that reduction of cell-bound IgE by omalizumab restores antiviral activity in asthma.

## 2. COVID-19 Pandemic Measures and RV Incidence

While PCR-based assays linking RV infection and asthma exacerbation are convincing, perhaps there has been no greater evidence of the link between respiratory viral infection and asthma exacerbation than the impact of COVID-19 on the incidence of asthma exacerbations in the United States and around the world. Following the initial surge of SARS-CoV-2 in March 2020, governmental agencies quickly developed a variety of preventative social restrictions in response to the mounting COVID-19 pandemic. At the height of the pandemic, remain-in-place orders and strict travel restrictions with enacted, nonessential personnel were encouraged to work remotely, large social gatherings and worship services were canceled, and educational institutions were closed. Since SARS-CoV-2 and RV are each transmitted via inhalation of respiratory droplets and aerosols, as well direct and indirect contact (fomites), the introduction and implementation of social distancing would be expected to “inadvertently” reduce the spread of RV and other respiratory viruses as well. Accordingly, the incidence of all non-COVID respiratory viruses decreased significantly during the height of strict COVID-19 pandemic measures, marked by social isolation and shelter-in-place orders.

For example, at Bambino Gesù Children’s Hospital in Rome, the number of positive respiratory pathogen panels in patients admitted with respiratory symptoms decreased by 80% during the 2020–2021 winter season compared to the previous two seasons, 2018–2019 and 2019–2020 [46]. The total number of RSV cases decreased from an average of 707 per year in 2018–2020 to five in 2020–2021. Similarly, influenza infections decreased from an average of 297 in 2018–2020 to none. RV detection also declined, although to a lesser extent; infections fell from 1030 and 1165 cases in 2018–2019 and 2019–2020 to 488 cases 2020–2021, a greater than 50% decline.

A similar analysis was performed at the Children’s Hospital of Philadelphia, with a particular focus on health care utilization during the pre-lockdown (1 January 2015–17 March 2020), lockdown (18 March 2020 to 5 June 2020) and phased reopening (6 June 2020 to 15 November 2020) time periods [47]. There was a significant decrease in RV, influenza A and B and RSV from the start of the lockdown through phased recovery, compared to historical rates, with minimal to no detection of influenza and RSV and markedly decreased detection of RV.

At the University of California-Davis, the positivity rate for respiratory pathogen panels between 25 March 2020 and 31 July 2020 decreased to 9.88 positive results per 100 tests compared to 29.90 positive results per 100 tests in the previous five years [48]. Influenza (93% reduction) and RV (81% reduction) fell significantly.

However, a closer look at the data reveals differences between the initial lockdown periods in the spring and summer of 2020, and the more relaxed protocols some countries employed in the fall of 2020. At that time, nurseries, kindergartens, primary schools, and many secondary schools had reopened. As noted above, school return has long been associated with RV infections [36]. Accordingly, in Finland, while a clear suppression of RV infection was seen between March and June of 2020 following the declaration of lockdown protocols, RV incidence rose to reference levels comparable to previous years shortly after the lockdown was lifted in June 2020 and later remained stable [49]. There was a similar trend for RV prevalence among 37 clinics and laboratories across Germany, with exceptionally strong RV suppression during strict social isolation protocols followed by a notable resurgence of RV infection in response to the relaxation of restrictions and reopening of schools and businesses [50]. In the United Kingdom, RV positivity declined following a national lockdown in late March of 2020 but rose sharply following reopening of state primary and secondary schools in early September 2020 [51]. In contrast to the resurgence of RV infections in the fall of 2020, the detection of other non-COVID respiratory viral infections including influenza, RSV, non-COVID coronaviruses and parainfluenza tended to remain low. Similar results were found in Brescia, Italy [52], France [53] Austria [54], New Zealand [55], South Korea [56], Japan [57] and, to a lesser extent, in the United States [58]. In some countries, the number of RV detections after the fall of 2020 surpassed that of earlier years. Data from New South Wales and Western Australia showed a significant increase in the detection of RV in 2020 relative to past years, while that of influenza and RSV were remarkably reduced [59].

Why was there a resurgence in RV compared to other respiratory viruses after the relaxation of social distancing measures? Due to its protein capsid, RV is more resistant to ethanol-containing disinfectant than viruses with a lipid envelope [60]. Consistent with this, in Japan, non-enveloped viruses with protein capsids (RV, human adenovirus, and coxsackievirus A and B) were detected to varying extents after relaxation of social distancing measures in contrast to the enveloped viruses (influenza virus, human parainfluenza virus 1-4, human metapneumovirus, and RSV) that were not [57]. Furthermore, perhaps due to its small size, facemasks are less effective in filtering out RV compared to influenza virus and coronavirus [61]. Finally, it has been noted previously that periodic declines in RV infection coincide with peak influenza virus activity [62], a process known as viral interference. Thus, the absence of influenza infections may have permitted an increase in RV. Indeed, a recent study from Israel examining daily incidence rates of infectious diseases during the first three months (April–June 2021) after the easing of social restrictions showed increases in the daily incidence rates of both respiratory and gastrointestinal infections in children ages 0–3 years [63], further evidence that lack of exposure to influenza and other infections may be associated with increased subsequent susceptibility to pathogens and disease.

## 3. Changes in RV-Induced Illness during the COVID-19 Pandemic

The disruption in normative transmission patterns of non-COVID respiratory viruses during the COVID-19 pandemic allowed a unique opportunity to analyze the effect of sociocultural factors on various viral-related illnesses, and the role of respiratory viruses in the pathogenesis of various diseases. A consortium of children’s hospitals in the United States compared the number of pediatric inpatient admissions for various diagnoses during the intensive COVID-19 lockdown period (15 March 2020 to 29 August 2020) compared to the same period in the previous three years [64]. There was a highly significant and sustained reduction in the number of pediatric inpatient admissions across all participating hospitals for asthma (81.3% reduction), bronchiolitis (80.1% reduction), pneumonia (71.4% reduction), and upper respiratory tract infections (68.9% reduction) during the COVID-19 period. In contrast, all other tracked illnesses (including malnutrition, anemia, hypovolemia, urinary tract infections, diabetes) experienced a transient decrease early in the COVID-19 pandemic and returned to pre-pandemic levels as overall inpatient admissions gradually recovered.

In the aforementioned Philadelphia study [47], reductions in RV, influenza A and B and RSV infections were accompanied a fall in asthma morbidity. During lockdown, the average number of weekly hospital encounters for asthma exacerbation decreased to 16% of pre-lockdown encounters. During phased reopening, the prevalence of acute care needs for asthma remained relatively reduced, as hospital encounters for asthma were only 40% of historical levels.

An observational study of 3100 subspecialist-treated adult patients with severe asthma in the U.S. showed significant reductions in exacerbations, emergency department visits and hospitalizations in April through August 2020 relative to the same months in 2019 [65]. Exacerbations remained lower than the prior year through May 2021.

A nationwide study of asthma hospitalizations in Japan showed a significant reduction in weeks 9 through 22 of 2020 compared to two earlier years [66]. In Kobe City, Japan, the typical spring increase in asthma exacerbations was not observed during the lockdown phase of the COVID-19 epidemic, but the fall peak was again observed after the state of emergency was lifted [67]. Hospitalizations for asthma exacerbations in early 2020 also decreased in Hong Kong [68] and Singapore [69]. In Hong Kong, the average admission count for adults with asthma decreased by 53.2%; this reduction was twice that for myocardial infarction, ischemic stroke, or gastric ulcer [68].

In perhaps the largest relevant study, a retrospective analysis of 100,362 asthma patients from across England showed substantial and persistent reduction in asthma exacerbations over the 18 months after the first COVID-19 lockdown [70]. Mean exacerbation rates (total number of exacerbation episodes per 100 patient-years) decreased from a range of 48.7–88.9 before the pandemic to a range of 23.2–34.1 during the pandemic.

While the most likely explanation for the substantial drop in asthma exacerbation rates seen during the pandemic is due to reduced exposure to respiratory viruses, other factors must also be considered. For example, the Hong Kong study [68] suggests the possibility that reductions in asthma hospitalization could have been due in part to the avoidance of visiting the hospital for fear of contracting a COVID-19 infection. However, a reduction in hospital admissions due to non-respiratory diseases was not seen in the U.S. children’s hospital study [64]. Since asthma exacerbations are a potentially fatal disease, it is reasonable to assume that families would have visited the hospital had their children developed severe asthma symptoms. Another factor which could have reduced asthma exacerbations during the lockdown phase of the pandemic is reduced exposure to outdoor pollution and allergens. In Kobe, Japan, there was a significant reduction in sulfur dioxide (SO_2_) levels during the state of emergency [67]. However, SO_2_ levels remained low when asthma exacerbations increased in the fall. The persistent reduction in asthma exacerbations after reopening of schools and businesses, in the face of increasing RV infections, is also poorly understood. Possibilities such as continued hygiene protocols and improved self-management have been proposed.

## 4. Does Asthma Increase Morbidity Due to SARS-CoV-2 Infection?

As we have seen above, overall the impact of the social distancing measures has been a reduction in asthma exacerbations and improvement in asthma control. The Pediatric Asthma in Real Life (PeARL) multinational cohort study reported improved asthma control in 65.9% of pediatric patients ages 4–18 years along with significant improvements in pre-bronchodilator FEV_1_ and peak expiratory flow rate during the early months of the pandemic (March to June 2020), compared to questionnaire responses and pulmonary function measurements recorded during the year prior [71].

However, in an individual patient, does the diagnosis of asthma increase the risk from COVID-19 infection? The preponderance of studies shows no increased risk of COVID-19 in asthma [72,73,74,75,76,77]. Certainly children with asthma are not at higher risk [78,79], based on their general resistance to severe COVID-19. Allergic inflammation has a protective effect [80]. Studies have shown that stimulation of differentiated human bronchial epithelial cells cultured at air-liquid interface with the type 2 pro-asthmatic cytokine IL-13 reduces SARS-CoV-2 viral replication [81] and decreases expression of ACE2, the SARS-CoV-2 receptor [82]. ACE2 expression is reduced in the nasal and airway epithelium of patients with allergic asthma [83,84]. On the other hand, the association between asthma and COVID-19 severity may be influenced by disease status and medication treatment, factors that have not been thoroughly evaluated in many studies. If there is an increased risk, this is likely driven by adult patients with uncontrolled or severe asthma, in particular non-allergic asthma [85,86,87,88]. Interestingly, while pro-allergic type 2 cytokines are associated with reduced ACE2 expression in asthmatic tissue, type 1 cytokines are associated with increased expression [89].

## 5. Interactions between RV and SARS-CoV-2

In this review, we have prevented evidence that social-distancing, restrictions on social gatherings, and increased hygiene protocols designed to prevent COVID-19 reduced the prevalence of RV and other respiratory viral infections, leading to a reduction in asthma morbidity. However, recent evidence suggests that prior infections with RV or other respiratory viruses can attenuate the severity of SARS-CoV-2 infection. RV-A16 blocks SARS-CoV-2 replication in cultured airway epithelial cells [90] and RV-A1A blocks SARS-CoV-2 replication in airway epithelial organoids [91]. Priming with RV-A1B protects mice against a lethal pulmonary infection with mouse hepatitis virus, a betacoronavirus that naturally infects the enteric tract of mice [92]. In a large academic Dutch hospital, SARS-CoV-2 infection was less common among employees who had received a previous influenza vaccination [93]. The protective effect of RV is dependent on the production of a type 1 IFN response, a process called viral interference. These results suggest the possibility that the increasing prevalence of RV could reduce the number of new COVID-19 cases [90].

## 6. Conclusions

Following the outbreak of COVID-19 in March 2020, social-distancing, restrictions on social gatherings, and increased hygiene protocols significantly reduced the opportunity for exposure to RV and other infectious agents. As a consequence, there was a significant reduction in both respiratory viral infections and asthma health care utilization. After school reopening in the fall of 2020, there was a specific increase in RV infections, in contrast to other respiratory viruses which remained diminished. School reopening was accompanied by a partial increase in asthma hospitalizations, but admissions remained lower than pre-COVID period. RV may be less susceptible to masking and disinfectants due to its small size and capsid coat. Just as family gatherings with young children drive asthma and COPD exacerbations in adults, younger children may be more likely to transmit SARS-CoV-2 infection to adults. Patients with allergic asthma do not suffer increased morbidity from SARS-CoV-2 infections, possibly due to the protective effects of IL-13 and other type 2 cytokines.

## Data Availability

Not applicable.

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
