# Peer review of "Effects of COVID-19 and Social Distancing on Rhinovirus Infections and Asthma Exacerbations"

_viruses, 2022, doi:10.3390/v14112340_

Round 1
Reviewer 1 Report
The manuscript from Kreger and Hershenson combines data from the University of Michigan Health with the data from the literature comparing pre- and post-implementation of COVID-19's social-distancing policies proposing a correlation between asthma and other lung diseases events with the circulation of respiratory viruses focusing mainly on the rhinovirus circulation.
The introductory section is clear and informative, describing essential features of rhinoviruses; however, I missed the presence of information mentioning existing treatments against the rhinoviruses.
The following sections describing the incidence of RV in different countries and populations of different ages draw a clear global-wide picture of the effect of the restrictions interfering with the incidence of respiratory viruses and respiratory diseases. Although it is very rich in information, I miss the presence of information mentioning the total number of hospital admissions within the observed period in the different studies aiming to prevent any bias due to the restrictions by themselves and/or fear of exposure to the SARS-CoV2.
The effect of inflammation markers impacting SARS-CoV2 infection is a very interesting addition to the manuscript; however, as this manuscript focuses on RV infection, the authors could, in addition, mention the new findings from Dee et al. (J Infect Dis. 2021 Jul 2;224(1):31-38. doi: 10.1093/infdis/jiab147), Cox et al. (Front Immunol. 2022 May 30;13:886611. doi: 10.3389/fimmu.2022.886611. eCollection 2022), or others that discuss the effect of pre-existing RV infection and the prevention of worsening of SARS-CoV2 infection.
Minor issues:
L26. For completeness, the authors could include a reference that provides more information about the rhinovirus study background.
L33. Reference 4 doesn’t describe the VP4 as an RNA anchor.
L47. The authors could include a study from a different group to corroborate their statement about RV virulence.
L50. Please adjust the references.
L198. The statement claiming RV is one of the main responsible for asthma exacerbations could be softened as more studies exclusively focusing on the correlation between RV infection and asthma are needed to understand the aetiology of this disease that have multiple know triggers.
L212. Please check the punctuation.
Figure: The authors could provide more information about the data collection to ensure reproducibility.
After correcting the abovementioned issues, I recommend the manuscript for publication in Viruses.
Author Response
The manuscript from Kreger and Hershenson combines data from the University of Michigan Health with the data from the literature comparing pre- and post-implementation of COVID-19's social-distancing policies proposing a correlation between asthma and other lung diseases events with the circulation of respiratory viruses focusing mainly on the rhinovirus circulation.
C1. The introductory section is clear and informative, describing essential features of rhinoviruses; however, I missed the presence of information mentioning existing treatments against the rhinoviruses.
R1. As the Reviewer knows there are no specific antiviral treatments for RV. However, new biologic therapies have been shown to decrease asthma exacerbations in children, most of which are triggered by respiratory viruses, dominated by RV. I have added a new paragraph in the first section (1. Rhinoviruses).
C2. The following sections describing the incidence of RV in different countries and populations of different ages draw a clear global-wide picture of the effect of the restrictions interfering with the incidence of respiratory viruses and respiratory diseases. Although it is very rich in information, I miss the presence of information mentioning the total number of hospital admissions within the observed period in the different studies aiming to prevent any bias due to the restrictions by themselves and/or fear of exposure to the SARS-CoV2.
R2. In the U.S children’s hospital study (Markham, et al. Inpatient use and outcomes at children's hospitals during the early COVID-19 pandemic. Pediatrics 2021), we already mention in the manuscript that there was no reduction in hospitalization for a number of unrelated non-respiratory illnesses such as malnutrition, anemia, hypovolemia, urinary tract infections, and diabetes.
However, we acknowledge that most studies did not look at hospitalization rates for other diseases, and it is conceivable that the observed reductions could in part have been due to avoidance of visiting the hospital for fear of contracting a COVID-19 infection. We added additional data on hospitalizations due to non-respiratory illnesses from the Hong Kong study to the revised manuscript, and now discuss other potential reasons for the observed reduction in asthma hospitalizations.
C3. The effect of inflammation markers impacting SARS-CoV2 infection is a very interesting addition to the manuscript; however, as this manuscript focuses on RV infection, the authors could, in addition, mention the new findings from Dee et al. (J Infect Dis. 2021 Jul 2;224(1):31-38. doi: 10.1093/infdis/jiab147), Cox et al. (Front Immunol. 2022 May 30;13:886611. doi: 10.3389/fimmu.2022.886611. eCollection 2022), or others that discuss the effect of pre-existing RV infection and the prevention of worsening of SARS-CoV2 infection.
R3. We agree this is a very interesting aspect of the story and have added a new paragraph on potential interactions between RV and SARS-CoV2 infections (new section 5. Interactions between RV and SARS-CoV2).
Minor issues:
C4. L26. For completeness, the authors could include a reference that provides more information about the rhinovirus study background.
R4. We have added two new references of the 1956-7 discovery of rhinoviruses.
C5. L33. Reference 4 doesn’t describe the VP4 as an RNA anchor.
R5. We agree, and in fact we decided it was more important to mention that VP4 forms a multimeric membrane pore, allowing extrusion of viral RNA genome into the cytoplasm. We deleted the previous references and added a new one (Panjwani et al.)
C6. L47. The authors could include a study from a different group to corroborate their statement about RV virulence.
R6. In addition to the Cox group, we had two additional references supporting the notion tht RV-C is more virulent than RV-A (Lauinger et al and Erkkola et al.). We added a third (Miller et al).
C7. L50. Please adjust the references.
R7. We apologize for this problem. We corrected the references.
C8. L198. The statement claiming RV is one of the main responsible for asthma exacerbations could be softened as more studies exclusively focusing on the correlation between RV infection and asthma are needed to understand the aetiology of this disease that have multiple know triggers.
R8. We understand this comment. We are probably a bit biased in this regard. As a pediatrician, my experience is that the majority of asthma attacks are caused by rhinovirus, depending on the time of year and prevalance of various respiratory viruses in the community. For example, one study detected on RV in 82% of all children admitted to an emergency room for acute asthma between January and July (Kling et al). I added this citation to the manuscript, but I also deleted the offending line above.
C9. L212. Please check the punctuation.
R9. We eliminated the extra period.
C10. Figure: The authors could provide more information about the data collection to ensure reproducibility.
R10. Based on the many questions and concerns about the Figure articulated by Reviewer 2, and the short turnaround time required for this response, I’m afraid we are forced to delete the Figure from the revised manuscript.
Reviewer 2 Report
Kreger and Hershenson presented a well-written and straight forward review based on the effects that COVID-19 and social distancing had on rhinovirus infection rates, and, consequently, to asthma exacerbations.
Respiratory viruses, and especially rhinovirus infections, are known to be responsible for exacerbating severe respiratory illnesses such as asthma or COPD, as demonstrated in several studies, some of which reported in this review. Similarly, statistics taken from healthcare providers in different countries, showed a significant decrease in asthma and or COPD exacerbation cases, starting from the spring of 2020, when the first lockdown measures were taken to contrast the SARS-CoV-2 spreading.
The reviewer therefore agrees with what is reported in the review. However, as a general comment, it is unclear whether the decrease in RV infections is the only cause for reduction of asthma/COPD exacerbation cases. Indeed, as discussed also in this study published in Lancet in June 2022 ("Impact of COVID-19 pandemic on asthma exacerbations: Retrospective cohort study of over 500,000 patients in a national English primary care database" by Shah et al.), the reduction in asthma/COPD exacerbation cases persisted long term, at least until November 2021. Reduction in respiratory diseases exacerbation cases is therefore lasting way longer than the heavy restrictions due to COVID-19 pandemic, and, most of all, continued even after RV infections raised to match pre-pandemic levels (as discussed in lines 114 to 157 of the review). Could the authors elaborate more on this topic, maybe indicating other possible causes why asthma exacerbation cases remained lower than the pre-pandemic?
As a minor comment, readability in line 36 to 41 is a bit unclear (it looks like some text got moved from one part to the other). Could the authors consider rephrasing?
Author Response
Kreger and Hershenson presented a well-written and straight forward review based on the effects that COVID-19 and social distancing had on rhinovirus infection rates, and, consequently, to asthma exacerbations.
Respiratory viruses, and especially rhinovirus infections, are known to be responsible for exacerbating severe respiratory illnesses such as asthma or COPD, as demonstrated in several studies, some of which reported in this review. Similarly, statistics taken from healthcare providers in different countries, showed a significant decrease in asthma and or COPD exacerbation cases, starting from the spring of 2020, when the first lockdown measures were taken to contrast the SARS-CoV-2 spreading.
C1. The reviewer therefore agrees with what is reported in the review. However, as a general comment, it is unclear whether the decrease in RV infections is the only cause for reduction of asthma/COPD exacerbation cases. Indeed, as discussed also in this study published in Lancet in June 2022 ("Impact of COVID-19 pandemic on asthma exacerbations: Retrospective cohort study of over 500,000 patients in a national English primary care database" by Shah et al.), the reduction in asthma/COPD exacerbation cases persisted long term, at least until November 2021. Reduction in respiratory diseases exacerbation cases is therefore lasting way longer than the heavy restrictions due to COVID-19 pandemic, and, most of all, continued even after RV infections raised to match pre-pandemic levels (as discussed in lines 114 to 157 of the review). Could the authors elaborate more on this topic, maybe indicating other possible causes why asthma exacerbation cases remained lower than the pre-pandemic? Need to read this and add comments: Shah SA, Quint JK, Sheikh A. Impact of COVID-19 pandemic on asthma exacerbations: Retrospective cohort study of over 500,000 patients in a national English primary care database. Lancet Reg Health Eur. 2022 Aug;19:100428.
R1. We agree with this comment. We added this important reference to the manuscript, and have also added an additional paragraph discussing other causes for the reduction in asthma exacerbations seen during and after the lockdown.
C2. As a minor comment, readability in line 36 to 41 is a bit unclear (it looks like some text got moved from one part to the other). Could the authors consider rephrasing?
R2. We have revised – we broke a long sentence into two separate ones.
Reviewer 3 Report
This review elegantly summarises the impact of social distancing on RV infection. Given that asthma exacerbation is highly driven by infection, they both correlate and are reduces during COVID. It would be useful to try to tease out data on non-infectious exacerbations and COVID, and investigate if COVID also impacted those exacerbation. The reviewer feels that this review should only tackle COVID and RV as data on asthma is poor or confounding.
Minor:
Most of the 2nd para of section 3 (line 173-178) should be included in section 2 (infection), rather than in section 3 (diseases).
The reviewer is unsure about including novel data in this review, as there is no method section or statement about ethic approval. What is included in RV infections (PCR dection at hospital)? On how many samples were those RV detected? Does it include RV-A, -B and -C?
Are those data publicly available? Also, limiting all the data to represent 9 months per year seems arbitrary. Maybe present monthly average or mention that data for 2022 is ongoing. The scale of Figure 1 is very misleading and should start at 0.
Is there more information of ACE2 expression in asthmatic versus non asthmatic? Or during exacerbation? Are allergic-asthmatic less likely or more likely to have COVID during allergy season?
Typo :
Line 37: « . » should be a coma.
Line 39: missing a full stop before The.
Line 50: references.
Line 195: should be Figure 1.
Author Response
This review elegantly summarises the impact of social distancing on RV infection. Given that asthma exacerbation is highly driven by infection, they both correlate and are reduces during COVID. It would be useful to try to tease out data on non-infectious exacerbations and COVID, and investigate if COVID also impacted those exacerbation.
C1. The reviewer feels that this review should only tackle COVID and RV as data on asthma is poor or confounding.
R1. We agree that the question of whether asthma is a risk factor for severe COVID-19 remains an area of active investigation. However, there is great interest in this topic. I would therefore like to include this section in the revised manuscript. I have added a sentence stating that “the association between asthma and COVID-19 severity may be influenced by disease status and medication treatment, factors that have not been thoroughly evaluated in many studies.” I have also added several new references on both sides of the issue.
C2. Most of the 2nd para of section 3 (line 173-178) should be included in section 2 (infection), rather than in section 3 (diseases).
R2. We shifted the first part of this paragraph to section 2, as suggested.
C3. The reviewer is unsure about including novel data in this review, as there is no method section or statement about ethic approval. What is included in RV infections (PCR dection at hospital)? On how many samples were those RV detected? Does it include RV-A, -B and -C? Are those data publicly available? Also, limiting all the data to represent 9 months per year seems arbitrary. Maybe present monthly average or mention that data for 2022 is ongoing. The scale of Figure 1 is very misleading and should start at 0.
R3. Based on the many questions and concerns about the Figure, and the short turnaround time required for this response, we are going to delete the Figure from the revised manuscript.
C4. Is there more information of ACE2 expression in asthmatic versus non asthmatic? Or during exacerbation? Are allergic-asthmatic less likely or more likely to have COVID during allergy season?
R4. In the original manuscript, we mentioned that ACE2 expression is decreased in asthma. I added an additional reference.
C5. Line 37: « . » should be a coma.
R5. We re-wrote this section.
C6. Line 39: missing a full stop before The.
R6. We re-wrote this section.
C7. Line 50: references.
R7. Thank you, I deleted the extra citations.
C8. Line 195: should be Figure 1.
R8. As noted above we deleted the figure from the revised manuscript.